# Phenotypic Characteristics, Antimicrobial Susceptibility and Virulence Genotype Features of *Trueperella pyogenes* Associated with Endometritis of Dairy Cows

**DOI:** 10.3390/ijms25073974

**Published:** 2024-04-03

**Authors:** Ning Liu, Qiang Shan, Xuan Wu, Le Xu, Yanan Li, Jiufeng Wang, Xue Wang, Yaohong Zhu

**Affiliations:** 1College of Veterinary Medicine, Northeast Agricultural University, Harbin 150001, China; nliu@neau.edu.cn (N.L.);; 2College of Veterinary Medicine, China Agricultural University, Beijing 100193, Chinajiufeng_wang@hotmail.com (J.W.); 3Key Laboratory of the Provincial Education, Department of Heilongjiang for Common Animal Disease Prevention and Treatment, College of Veterinary Medicine, Northeast Agricultural University, Harbin 150030, China

**Keywords:** *Trueperella pyogenes*, drug resistance, virulence gene, endometritis, antibiotic

## Abstract

*Trueperella pyogenes* can cause various infections in the organs and tissues of different livestock (including pigs, cows, goats, and sheep), including mastitis, endometritis, pneumonia, or abscesses. Moreover, diseases induced by *T. pyogenes* cause significant economic losses in animal husbandry. In recent large-scale investigations, *T. pyogenes* has been identified as one of the main pathogens causing endometritis in lactating cows. However, the main treatment for the above-mentioned diseases is still currently antibiotic therapy. Understanding the impact of endometritis associated with *T. pyogenes* on the fertility of cows can help optimize antibiotic treatment for uterine diseases, thereby strategically concentrating the use of antimicrobials on the most severe cases. Therefore, it is particularly important to continuously monitor the prevalence of *T. pyogenes* and test its drug resistance. This study compared the uterine microbiota of healthy cows and endometritis cows in different cattle farms, investigated the prevalence of *T. pyogenes*, evaluated the genetic characteristics and population structure of isolated strains, and determined the virulence genes and drug resistance characteristics of *T. pyogenes*. An amount of 186 dairy cows were involved in this study and 23 *T. pyogenes* strains were isolated and identified from the uterine lavage fluid of dairy cows with or without endometritis.

## 1. Introduction

Dairy cow endometritis is a common disease in the dairy farming industry, one which often causes a decline in the milk production and reproductive performance of dairy cows and thus brings serious economic losses to cattle farms [1]. After calving, dairy cows are more likely to suffer from endometritis, which affects the maintenance and establishment of pregnancy and thereby reduces their fertility [2]. Uterine diseases are commonly associated with *Escherichia coli*, *Arcanobacterium pyogenes*, *Trueperella pyogenes*, *Fusobacterium necrophorum*, and *Prevotella* [3]. Significantly, *T. pyogenes* has a high isolation rate in dairy cows suffering from endometritis in China, which is often aggravated infection with pathogenic bacteria such as *Escherichia coli*, *Streptococcus*, and *Staphylococcus* species [4].

*T. pyogenes* is a Gram-positive, polymorphic, non-spore forming, non-motile, non-capsule, facultative anaerobic bacterium characterized by its fermentation metabolism and strong proteolytic activity [5]. It is a causative agent of metritis, abortion, mastitis, infertility and pneumonia in dairy herds [6]. Most of the published data pertain to the infection of *T. pyogenes* in edible animals from Europe, China, Japan, Brazil, and the United States [7,8,9,10,11,12,13,14,15]. In livestock, *T. pyogenes* infections can cause significant economic losses, mainly in breeding cattle and pigs, resulting in reduced meat and milk production, decreased reproductive efficiency, and sometimes the need to cull diseased animals [16].

Several virulence factors are significant contributors to the pathogenicity expressed by *T. pyogenes* [17]. Pyolysin (PLO) is encoded by the *plo* gene, a hemolysin with cytolytic activity on immune cells [18]. Pyolysin, a cholesterol-dependent cytolysin (CDCs), is a water-soluble monomer forming transmembrane β-barrel channels in the cell membrane by binding cholesterol [19]. *nanH*, *nanP*, *cbpA*, *fimA*, *fimC*, *fimG*, and *fimE* are other virulence genes which play key roles in the pathogenicity of *T. pyogenes*, such as in their adherence to epithelial cells and to collagen rich tissues [20,21]. Based on current knowledge, all pathogenic *T. pyogenes*, isolated from within a wide range, secrete *plo*. However, *nanP*, *cbpA*, and *fimA* are only expressed in some isolates [17].

Aminoglycosides, β-lactam antibiotics, tetracyclines, macrolides, and fluoroquinolones are antimicrobials commonly treated with different bacterial infections, including *T. pyogenes* [13,22]. The antibiotic resistance of *T. pyogenes* has become a new and serious issue due to the widespread use of antibiotics in agriculture. It has also brought great difficulties to clinical treatment and poses a serious security threat to human life. [11,12,23]. Although we have known about *T. pyogenes* for a long time, little is known about molecular and antibiotic resistance-based properties of *T. pyogenes* in the cases of endometritis and metritis [6]. The consequences will be serious if the application of antibiotics is invalid [1].

Altogether, monitoring pathogenic microorganisms, such as *T. pyogenes*, is a vital link in the control and reduction of their hazards to animals and humans. Our objectives were to compare the uterine flora of healthy cows and endometritis cows, in order to investigate the prevalence of *T. pyogenes* within them and to assess the genetic characteristics and population structure of *T. pyogenes* isolates, and to identify the virulence genes and drug resistance characteristics of *T. pyogenes.*

## 2. Results

### 2.1. Clinical Sampling of T. pyogenes from the Uterus of Dairy Cows

Dairy cows with polymorphonuclear neutrophils (PMN) ≥ 18% (*n* = 87) were judged to have cow endometritis. Those for which the uterine lavage fluid was accompanied by a yellow or reddish-brown purulent mucus were scored mainly as 2 or 3, while a number scored 1 (Figure 1A). Dairy cows with PMN < 18% were classified as healthy cows, their lavage fluid appeared clear or translucent and scored 0 points. The types of colonies found on sheep blood plates that were coated with the uterine lavage fluid of healthy cows were relatively simple, while a number of different types of colonies were found on sheep blood plates that were coated with the uterine lavage fluid of cows with endometritis (Figure 1B). All isolated bacteria were identified using 16 s DNA sequencing. In the uterine lavage fluid cytological examination of healthy cows (Figure 1C(a,b)), the background was clean with only few large endometrial epithelial cells that were shed (black arrow). These are integral in shape with round, pink cytoplasm and purple nucleus. There were no neutrophils found. However, in Figure 1C(c–h), one can see that the uterine lavage fluid of cows with endometritis has a large number of cells and that the overall visual field is darker. The cellular components are mainly neutrophils (white arrows). The nucleus is stained blue and is clearly lobed or horseshoe shaped. Occasionally, a small amount of endometrial epithelial cells and red blood cells with incomplete cell membrane structures appeared.

A total number of 186 dairy cows was involved in this study, from three different dairy farms in China. The uterine lavage fluids of 186 cows 3 to 4 weeks postpartum were labeled from 1 to 186 in the order that they were sampled and preliminary diagnosis was made based on the clinical symptoms and the uterine lavage fluid scores. Table 1 outlines a total of 99 healthy cows and 87 dairy cows with endometritis. The isolation rate of *T. pyogenes* from healthy dairy cows is only 2%, which is much lower than that from dairy cows with endometritis (24.1%).

### 2.2. The Differences of Uterine Microbiota of Healthy Dairy Cows and Dairy Cows with Endometritis

In this study, the16 s DNA sequencing was performed on all isolated bacteria from all 186 dairy cows. The NCBI’s BLAST system was used for homology comparison analysis of clinically isolated strains. In Figure 2, the number of common pathogenic bacteria isolated from healthy cows is significantly less than that of cows with endometritis, especially *Bacillus cereus* and *T. pyogenes*. In endometritis cows, *B. cereus*, *T. pyogenes*, *Streptococcus* and *Staphylococcus* sp. become the dominant isolated flora. *Arthrobacter* sp., *Lactobacillus* sp., *B. subtilis*, *B. licheniformis*, *Coynebaterium* sp., and *Pseudomonas aeruginosa* have an absolute advantage in the isolation rate of healthy dairy cows.

### 2.3. Growth Characteristics of T. pyogenes

The growth characteristics of clinically isolated *T. pyogenes* are shown in Figure 3. Purified needle-like white *T. pyogenes* colonies with obvious hemolytic rings around them were seen with 24 h incubation on sheep blood plates (Figure 3A). The size of *T. pyogenes* under scanning electron microscope is usually 1.0–1.5 μm × 3–8 μm and it is typically short and rod shaped (Figure 3B).

The evolutionary trees of different isolated *T. pyogenes* strains are shown in Figure 4A. It can be seen that most species of *T. pyogenes* are closely related. It is clear from the results that the genetic relationship between *T. pyogenes* strains isolated from the same cattle farm is relatively similar. The growth curves of all 23 isolated and purified clinical strains of *T. pyogenes* were measured (Figure 4B). The growth rates of all isolated *T. pyogenes* strains were not exactly the same and most of the strains reached the logarithmic growth phase between 9 and 12 h.

### 2.4. Detection of Antimicrobial Susceptibility Profiles on T. pyogenes Strains

Antimicrobial susceptibility testing (Table 2) was also performed to better explore the characteristics of *T. pyogenes*. It is clear that multidrug resistance has appeared in isolated *T. pyogenes* strains. More than 33% of isolated strains are resistant to 7 or more antibiotics (Figure 5), while more than 83% of isolated strains in this study are resistant to three or more antibiotics. The current situation is very severe and requires more research on antibiotic abuse and on alternatives to antibiotics.

### 2.5. Detection of Virulence Genes and Their Pathogenicity to Endometrial Epithelial Cells

The quantitative measurement of lactate dehydrogenase method was used to determine the cell mortality after a 9 h challenge of *T. pyogenes* (Figure 6). The cell death rates upon infection by TP1804, TP1901, and TP1908 were particularly higher than other strains. The virulence genes were also tested and are shown in Figure 7. Among the tested virulence genes, *plo* was 100% detected, followed by *fimA* (93.8%) and then *nanP* (82.9%), *fimC* (74.3%), *fimE* (71.8%), and *nanH* (69.2%). These virulence genes play important roles in the growth, adhesion, and toxicity of *T. pyogenes* and are the reason that TP1804, TP1901, and TP1908 cause such high cell mortality.

## 3. Discussion

The novelty, large sample size of dairy cows, and multiple analyses are the advantages of this study. The type and quantity of bacteria present in uterine lavage fluid may have a certain relationship with the severity of endometritis and the development of prognosis. Based on the results of this study, we can determine that the types of pathogenic bacteria isolated from the uterine lavage fluid of a single cow with endometritis are far more diverse than those of healthy cows, especially *B. cereus* and *T. pyogenes,* which have much higher separation rates. The uterine flora of cows suffering from endometritis is disordered and the flora is more complex. The bacterial abundance in cows with endometritis and healthy cows is distinctive. In healthy cows, *B. cereus* and *T. pyogenes* are rarely separated and there is no separation of *Helcococcus* and *Peptoniphilus sallinarum*. Moreover, the uterine microbiota of healthy cows was characterized by an enrichment of probiotics—*Lactobacillus* [4]. Apart from bacterial infections, other factors affecting the incidence rate of cow endometritis include herd size, population density, feeding environment and treatment methods.

*T. pyogenes* is a pathogenic bacterium that is particularly well associated with endometritis [24,25]. In cows with endometritis, *T. pyogenes* and *B. cereus* have a remarkably higher isolation rate than other pathogenic microorganisms. In this experiment, only two *T. pyogenes* strains were isolated from healthy dairy cows, especially in the Beijing area without isolation. The isolation rate of *T. pyogenes* in cows with endometritis in the Beijing area is as high as 28.6%. One cannot say with certainty that *T. pyogenes* is the cause of endometritis; however, it cannot be ignored that *T. pyogenes* plays a vital role in its occurrence and development. Endometritis has brought huge economic losses to dairy farms and also has an impact on the conception of dairy cows. However, the treatment of endometritis accompanied by *T. pyogenes* infection is more difficult and recurrent infections often occur.

Bacterial infections in the uterus and vagina after postpartum can cause postpartum microbial disorders and affect their evolution, resulting in disease for the dairy cows [11,24]. This may be because *T. pyogenes* contributes to the destruction of the environment in the uterus and provides conditions for infection by other pathogenic bacteria. Our results show that *T. pyogenes* is positively related to clinical endometritis and likely to work synergistically with *Bacillus*, *Staphylococcus*, *Streptococcus* and other bacteria to cause the disorder of the uterine microbiota of endometritis dairy cows. It is therefore plausible that uterine pathogens might assist each other in avoiding uterine defense mechanisms and interact to facilitate colonization of the endometrium [4]. This disorder is regarded as an important indicator of uterine infection in dairy. Therefore, the role of *T. pyogenes* in dairy cow uterine diseases and the synergistic effect of pathogenic bacteria cannot be underestimated and more research should be undertaken.

Although there have been many reports of *T. pyogenes* being associated with endometritis all over the world, there are still few reports on the mechanism of the emergence of multidrug resistant *T. pyogenes* strains. In this study, more than 80% of clinical isolates are resistant to AMC and over 60% of clinical isolates are resistant to TET. Furthermore, 83.3% of the strains have resistance to three or more antibiotics. The isolates of *T. pyogenes* were all multidrug resistant in this study. Even more frightening is that TP1806 is resistant to all ten antibiotics used in this study. This indicates that antibiotics are used irregularly and for a long time during the feeding process. Although humans have used antibiotics for a long period and antibiotics have brought huge safety risks to human and animal health, antibiotic treatment is still the primary method of treating bacterial infections. Bacterial resistance monitoring can provide a theoretical basis and guidance for clinical treatment medication. Though the isolation rate of *T. pyogenes* is very high, it has received little attention in the current investigation of the dysbacteriosis in the postpartum uterus of dairy cows. Exhaustive investigation seeking to establish a systematic knowledge of bacterial resistance should be taken seriously and as soon as possible. The recurrent infections and the difficulty of thoroughly eliminating infections in the same cattle farms are even more severe.

The infections caused by *T. pyogenes* in both domestic and wild animals worldwide are opportunistic in nature and adverse environmental and host-related factors play a relevant role in the establishment of disease [5,15,26,27]. Presentation of putative and known virulence genes are significant in the determination of the pathogenicity of the *T. pyogenesis* isolates [28]. Virulence genes, including *nanH*, *nanP*, *plo*, *fimA*, *fimC*, and *fimE*, were all detected in the *T. pyogenes* strains isolated in this study. However, only the *plo* gene was found in all isolates. The expression of the virulence gene *fimA* is also as high as 93.8%, indicating that its role in the pathogenicity of *T. pyogenes* cannot be ignored. The carrier rates of other virulence genes were excavated in a very different manner. The expression of virulence genes plays an important role in the pathogenicity and biofilm formation of *T. pyogenes* [29]. However, the establishment and development of the *T. pyogenes* infection may not only derive from known virulence factors, but also from other unknown bacterial factors [5]. Our data indicate that the synergy of bacteria may be an important part of the pathogenesis. More investigation may be needed to study the virulence genes of *T. pyogenes.*

## 4. Materials and Methods

### 4.1. Biosecurity Statement

Clinical isolates of *T. pyogenes* were treated strictly in accordance with the state council of the people’s republic of China, and its regulations on the biological safety of pathogen microbiology laboratories (000014349/2004-00195). We conducted all of the necessary safety operations in order to avoid pathogen transmission and infection.

### 4.2. Experimental Design and Sample Collection

The sample collection of this research was executed on three commercial dairy farms in Heilongjiang, Beijing, and Hebei, the herds consisted of 3000, 800, and 500 milking Holstein cows, respectively. Amounts of 65, 74, and 47 dairy cows were recruited from Heilongjiang, Beijing, and Hebei, respectively. The cows selected for this study showed the following clinical symptoms of endometritis: retained placenta, purulent mucus at the vaginal opening, depressed mental state, and poor uterine involution. Based on the medical history and medication of cows born within 30 days of calving during the sampling period, this experiment selected cows without diseases such as mastitis, ketosis, hoof disease, true stomach displacement, postpartum paralysis, and with a body temperature below 39.5 ° C. None of the selected 186 dairy cows were supplied for an antibiotic treatment before the sample collection. The uterine lavage fluid of all cows, of which the average parity was about three, was collected three to four weeks after calving. Between three and four weeks of delivery, all cows were subjected to vaginal examination, rectal palpation, and endometrial cytology. Their other clinical symptoms were also recorded. Vaginal discharge was scored, as previously described, from 0 to 5 [30]. Dairy cows were classified as having clinical endometritis as previously described [4]. Dairy cows with a proportion of PMN ≥ 18% by cytological examination were classified as having subclinical endometritis in the absence of purulent vaginal discharge. Dairy cows with a clear or translucent vaginal discharge without putrescence and with a proportion of PMN < 18% were classified as healthy.

Uterine flush samples were collected, as previously mentioned [31]. Between three and four weeks of delivery, dairy cows were chosen to have their uterus flushed with saline using a stainless-steel uterine infusion pipette and an inflated balloon. Briefly, each cow was fixed before sampling and the perineum was disinfected with 75% alcohol. A lavage hose on the outside of the pipette was inserted into the cervix. A cleaned 20 mL syringe was used to inflate the air bag with air at the top of the pipette in order to prevent contamination and hose slippage. The uterus was injected with 20–30 mL sterilized warm normal saline and the liquid sample was drawn after gentle agitation. The uterine lavage fluid was sealed in a sterile 50 mL tube and, within 3 h, was transported on ice to the laboratory for subsequent processing.

### 4.3. Cytological Examination

After centrifuging the collected uterine lavage fluid at 3000× *g* for 10 min, a small amount of precipitate was drawn and re-suspended in sterile normal saline. A Pasteur tube was used to suck a small amount of re-suspension and drop it on one end of a glass slide. After smearing with another glass slide, the cell smear was dried and stained with the Diff-Quick method. The cytological examination was observed under a microscope by ×400 magnification in order to measure the proportion of PMN. Dairy cows with a proportion of PMN < 18% were considered healthy cows, while those with a proportion of PMN ≥ 18% were considered diseased cows.

### 4.4. Clinical Isolates of T. pyogenes Culture and Sequencing

The uterine lavage fluid was evenly vortexed and diluted with normal saline 10 times. The original lavage fluid solution and the dilution were evenly spread on 5% sheep blood plates which were incubated upside down in a 37 °C incubator for 24 h. Each concentration was duplicated. Preliminary screening was carried out according to the characteristics of the colony morphology and the presence of hemolysis loops. A single colony was cultivated on a new blood plate using an inoculating loop with the three-dimension line method. After incubation in a 37 °C incubator for 24 h, a purified single colony was incubated in brain–heart infusion (BHI) broth (Aobox, Beijing, China) for 24 h until the bacterial liquid became turbid. Then, the bacteria were separated by centrifugation (10,000× *g*, 5 min), followed by three sterile saline washings, 3% glutaraldehyde fixation, and ethanol and drying agent dehydration. The bacteria were finally coated with gold and observed under a Hitachi SU-8010U scanning election microscope (Hitachi, Tokyo, Japan)

The bacteria were cultured overnight, and the DNA was extracted with a TIANamp bacterial DNA kit (DP302, TIANGEN, Beijing, China). The PCR (Applied biosystems, 9902, Carlsbad, CA, USA) amplification was performed using 16S gene primers (F: AACTGGAGGAAGGTGGGGAT, R: AGGAGGTGATCCAACCGCA). The PCR products were sent to Beijing Tianyi Huiyuan Biotechnology Co., Ltd. for sequencing. The sequencing results were compared and analyzed using BLAST program (National Center for Biotechnology Information, Bethesda, MD, USA). In this study, 23 strains of *T. pyogenes* were isolated from uterine lavage fluid.

### 4.5. Determination of Growth Conditions

To examine different growth rates of all clinical isolates of *T. pyogenes* and the standard strain, ATCC 19411, bacterial growth curves were measured with a turbidity meter for all of 24 strains of *T. pyogenes* (BioMérieux, Durham, NC, USA). Sterilized turbidity tubes (Solarbio, Beijing, China) were filled with 1 mL BHI broth and a bacterial solution was dropped until its turbidity reached 0.5 Mcfarland standard (MCF). BHI broth (198 μL) was added to each well of a 96-well plate and then 2 μL of bacterial solution with a turbidity of 0.5 MCF was added to each well. Each *T. pyogenes* strain was repeated three times. The plate was placed in a microplate reader (Tecan, Männedorf, Switzerland) at 37 °C and measured OD_600_ every hour for a continuous 27 h measurement.

### 4.6. Determination of Minimum Inhibitory Concentration (MIC)

The MIC of *T. pyogenes* was determined by the broth microdilution method and *Escherichia coli* ATCC 25922 was used as quality control reference. The test was operated strictly according to performance standards for antimicrobial susceptibility testing M100-S28 of the clinical and laboratory standards institute (CLSI). Briefly, antibiotics were twofold diluted in BHI, and then mixed with bacterial suspensions (1.0 × 10^6^ cfu/mL) in an equal volume. The mixture was incubated in a 96-well plate at 37 °C for 18 h. The MIC was defined as the minimum concentration to inhibit bacterial growth. The antibiotics used in this study were as follows: azithromycin (AZM), cefazolin (CZ), ciprofloxacin (CIP), enrofloxacin (ENR), streptomycin (STR), amoxicillin (AMC), gentamicin (GM), kanamycin (KAN), ampicillin (AMP), and tetracycline (TET). Three independent repetitions per trial were conducted. The isolates were defined as “susceptible”, “intermediate”, or “resistant” based on the MICs for each antimicrobial agent.

### 4.7. Lactate Dehydrogenase (LDH) Assay

The death of cells (2 × 10^5^ cells/well) with different treatments was evaluated with an LDH Cytotoxicity Assay (Beyotime, Beijing, China) following the instructions of manufacturer [1]. The results were calculated with the following formula: % cytotoxicity = (infected LDH − control LDH)/(max lysis LDH − control LDH) × 100%. Qualitative data were expressed as mean ± standard error of mean value (SEM; *n* = 3 or 6).

### 4.8. Virulence Gene Testing

After the bacterial DNA was extracted according to the above method, the virulence gene primers (Table 1) were used for virulence gene detection. PCR products were analyzed with 1% agarose gel electrophoresis for 30 min at 125 V in 1× Tris-acetate-EDTA buffer, stained with ethidium bromide and photographed under UV trans-illumination.

### 4.9. Data Analysis

All of the result graphs in this article were created using Graphpad Prism version 9 (GraphPad Software, Boston, MA, USA) [32]. The phylogenetic analysis between isolates of the *T. pyogenes* structure was evaluated using MEGA X: Molecular Evolutionary Genetics Analysis across Computing Platforms, for sequence alignments and for the inference of evolutionary trees.

## 5. Conclusions

In conclusion, *T. pyogenes* infection in Chinese dairy farms is induced by isolates possessing distinct phenotypes and virulence genotypes. However, there is no evidence to suggest a strong correlation between the prevalence of *T. pyogenes* and the virulence genes it expresses. This study selected 186 dairy cows for bacterial testing through rigorous screening. Among the 23 isolated strains of *T. pyogenes*, more than 83.3% are multidrug-resistant bacteria and all of them also carry multiple virulence genes. We propose that, if the emergence of multidrug resistant *T. pyogenes* cannot be effectively and safely prevented and treated, the threat to food safety and even human health will pose a serious challenge to animal husbandry. This study provides a theoretical basis for future research on endometritis.

## Figures and Tables

**Figure 1 ijms-25-03974-f001:**
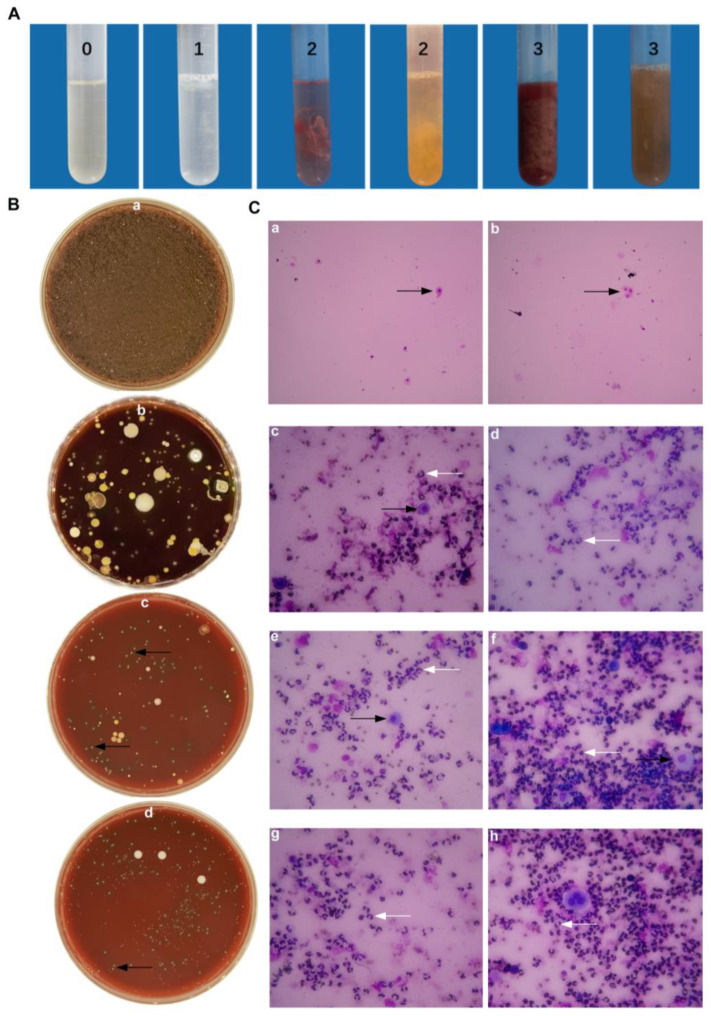
Epidemiological investigation on the collection of uterine lavage fluid in cattle farms. (**A**) Uterine lavage fluid grading standard from 0 to 3 points, indicating increasing severity. (**B**) The growth of bacteria from uterine lavage fluid on the blood plate from healthy cows (**B**(**a**)) and endometritis cows (**B**(**b**)). (**B**(**c**,**d**)) Black arrows are pointed at *T. pyogenes,* with transparent hemolysis rings. (**C**(**a**,**b**)) Images from health cows. (**C**(**c**–**h**)) Images from endometritis cows. White arrows indicate PMN; black arrows indicate BEECs.

**Figure 2 ijms-25-03974-f002:**
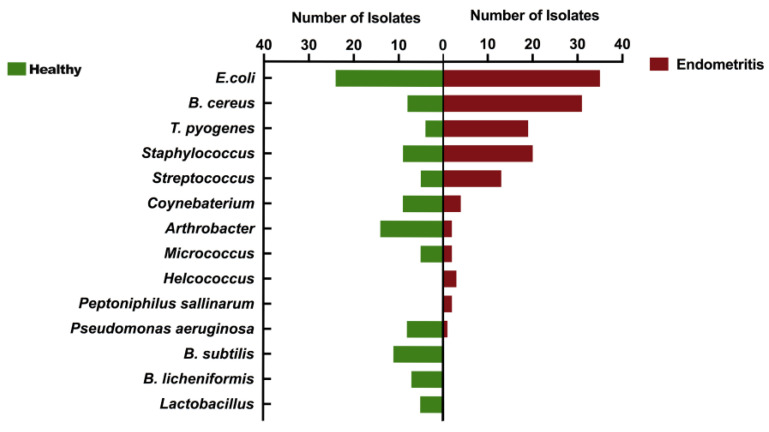
Comparison of uterine microbiota of dairy cows with or without endometritis. Comparison indicates the types of dominant flora in the uterus of healthy cows and endometritis cows.

**Figure 3 ijms-25-03974-f003:**
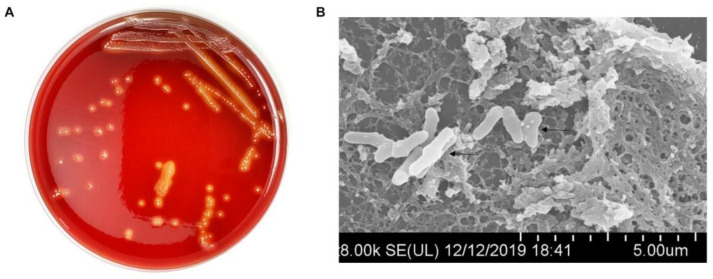
Exploration of the growth characteristics and cytotoxicity of *T. pyogenes*. (**A**) The phenotype of *T. pyogenes* on the sheep blood plate. (**B**) *T. pyogenes* observed under scanning electron microscope.

**Figure 4 ijms-25-03974-f004:**
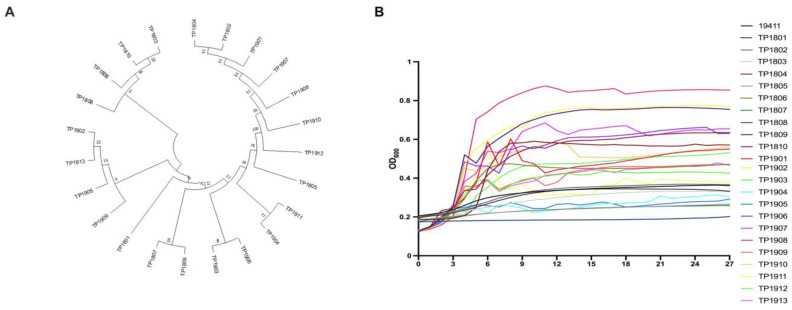
Detection of the genetics and growth characteristics of *T. pyogenes*. (**A**) The evolutionary tree shows the phylogeny of clinical isolates of *T. pyogenes*. (**B**) The growth curves of all isolated *T. pyogenes* for 27 h.

**Figure 5 ijms-25-03974-f005:**
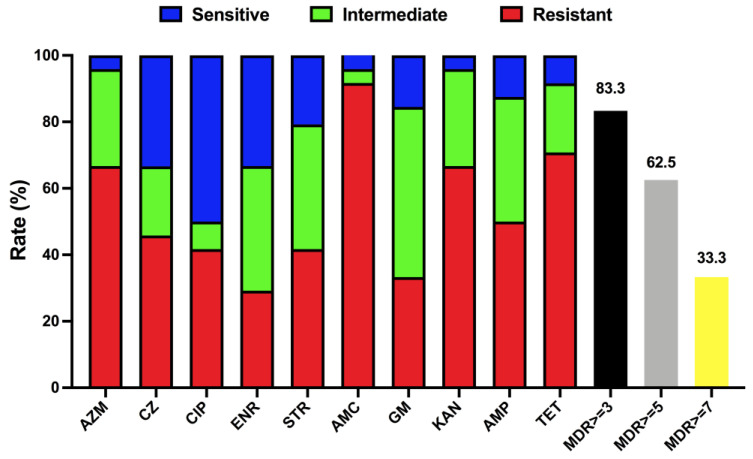
Antimicrobial susceptibility of *T. pyogenes* isolated from uterine lavage fluid in China. Blue, green, and red bars represent the proportion of sensitive strains, moderately resistant strains, and resistant strains, respectively. Black, yellow, and grey bars represent the respective strain’s proportion of multidrug resistance (MDR ≥ 3, 5, and 7).

**Figure 6 ijms-25-03974-f006:**
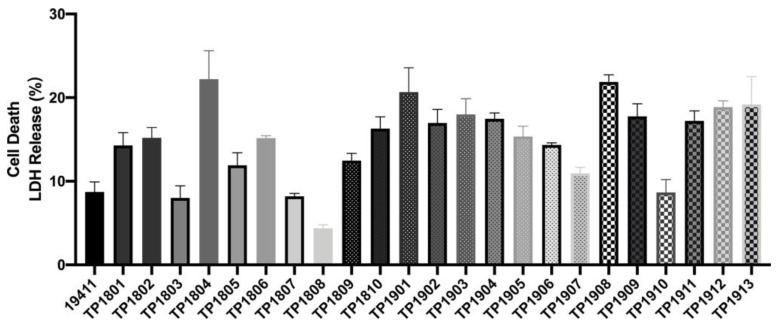
Cytotoxicity of *T. pyogenes* on endometrial epithelial cells.

**Figure 7 ijms-25-03974-f007:**
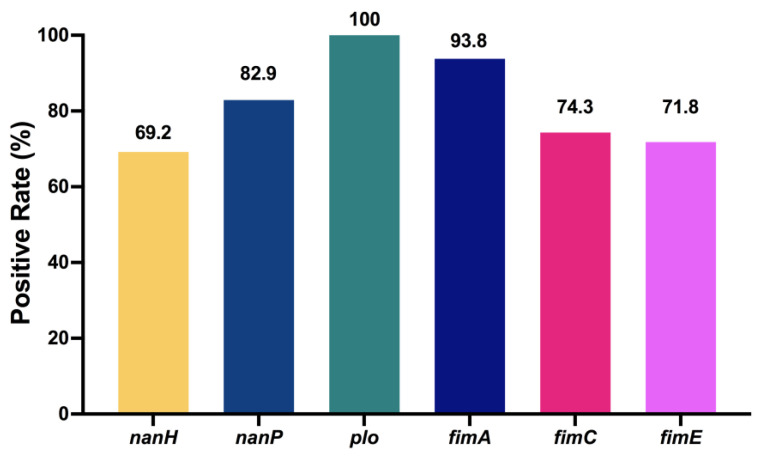
Detection rate of virulence genes in *T. pyogenes* from uterine lavage fluid in China. The numbers at the top of the bars indicate the positive rate of virulence genes.

**Table 1 ijms-25-03974-t001:** Isolation rates of *T. pyogenes* strains from different dairy farms.

Sources	Sample Numbers		*T. pyogenes* Numbers		Isolation Rates (%)	
	Healthy Cows	Diseased Cows	Healthy Cows	Diseased Cows	Healthy Cows	Diseased Cows
Heilongjiang	35	30	1	7	2.9	23.3
Beijing	39	35	0	10	0	28.6
Hebei	25	22	1	4	4	18.2
Total	99	87	2	21	2	24.1
	186		23		12.4	

**Table 2 ijms-25-03974-t002:** Comparisons of minimum inhibitory concentrations range (MIC) of *T. pyogenes* from different dairy farms.

Strain	Antibiotics									
	AZM	CZ	CIP	ENR	STR	AMC	GM	KAN	AMP	TET
ATCC 19411	R	I	S	S	I	R	R	R	S	R
TP1801	R	R	R	R	R	R	I	R	R	R
TP1802	R	S	S	I	R	R	R	R	R	R
TP1803	R	R	R	R	I	R	I	I	R	I
TP1804	R	R	S	I	R	R	R	R	R	R
TP1805	I	S	R	I	R	I	S	S	I	I
TP1806	R	R	R	R	R	R	R	R	R	R
TP1807	I	S	S	S	I	S	S	I	S	S
TP1808	R	R	R	R	I	R	R	R	R	R
TP1809	I	I	S	S	I	R	R	R	I	I
TP1810	R	R	R	R	R	R	I	R	R	R
TP1901	I	R	S	I	S	R	I	I	R	R
TP1902	I	S	S	S	S	R	S	I	S	R
TP1903	R	S	S	I	I	R	I	R	I	R
TP1904	R	I	R	I	I	R	I	R	I	R
TP1905	I	R	S	S	R	R	I	I	R	R
TP1906	R	I	R	I	I	R	I	R	R	R
TP1907	R	R	S	R	R	R	R	I	R	R
TP1908	R	S	S	S	S	R	I	R	I	R
TP1909	R	R	R	R	R	R	I	R	I	R
TP1910	I	R	S	S	I	R	I	R	R	I
TP1911	S	I	S	S	R	R	I	R	I	R
TP1912	R	S	I	I	S	R	R	R	I	I
TP1913	R	S	I	I	S	R	I	I	I	S

AZM, Azithromycin; CZ, Cefazolin; CIP, Ciprofloxacin; ENR, Enrofloxacin; STR, Streptomycin; AMC, Amoxicillin; GM, Gentamicin; KAN, Kanamycin; AMP, Ampicillin; TET, tetracycline.

## Data Availability

Data is contained within the article.

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
