# Peer review of "Phenotypic Characteristics, Antimicrobial Susceptibility and Virulence Genotype Features of *Trueperella pyogenes* Associated with Endometritis of Dairy Cows"

_ijms, 2024, doi:10.3390/ijms25073974_

Round 1
Reviewer 1 Report
Comments and Suggestions for Authors
The authors of the manuscript addressed the important issue of assessing the impact of T.pyogenes on the occurrence of endometritis in dairy cows. This microorganism, due to the presence of numerous virulence genes in its genome and synergistic action with other microorganisms, may influence the development of endometritis.
The work indicates the need for further observations and research on the prevalence of T.pyogenes in dairy herds.
The authors conducted a study on samples from cows from 3 herds. These herds varied in size (quite significantly). On what basis were the animals from which samples were taken selected? Despite significant differences in the size of the orchards, the number of samples from each of them did not vary to such a large extent. Do the authors believe that the herd size had an impact on the incidence of endometritis? What could cause these differences?
In the "Discussion", it would be worth devoting some attention to comparing the occurrence of virulence genes in the genome of strains isolated from healthy cows and those with endometritis. Were statistically significant differences observed?
Author Response
Dear Reviewer:
Thank you so much for your suggestions. We have revised the manuscript and uploaded our responses within the word document.

Reviewer 2 Report
Comments and Suggestions for Authors
The authors present an exploratory study of healthy and diseased cows regarding the bacterium Trueperella pyogenes, which may be zoonotic. The study highlights the complexity of uterine infections in dairy cows and highlights the important role of T. pyogenes and the pressing problem of antibiotic resistance. The study is well conducted and the results are presented in a very interesting way. However, there are some discrepancies as follows:
Please enter all species names in italics.
Lines 29-56: Please remove repetitions of arguments.
Line 85: Please provide the full abbreviation.
Line 254: Please provide the clinical symptoms in detail.
Subheading 4.7.: Did you perform this process manually or through an automated system? Please provide details.
Line 133: You didn't mention electron microscopy in the methodology?
No ANOVA results were presented.
Overall, the impression is that each section e.g. introduction, methodology, results and discussion were written by different authors and were not coordinated before the manuscript was put into final form. Because no statistical analysis was included in the manuscript, the conclusions presented could not be confirmed. In this scenario, I will recommend major revisions and give authors an opportunity to improve the presentation of their interesting work to the audience.
Comments on the Quality of English LanguageMinor English corrections.
Author Response
Dear Reviewer:
Thank you so much for your suggestions and another chance to revise the manuscript. We have revised the manuscript and uploaded our response within the word document.

Round 2
Reviewer 2 Report
Comments and Suggestions for Authors
Where are the statistical results as mentioned in the methodology and conclusions?
Comments on the Quality of English LanguageMinor revisions.
Author Response
Dear Reviewer,
Thank you so much for your concern. In fact, this was a mistake we made unintentionally. We carelessly thought that Figure 6 in the article had statistics, but we did not provide them. In fact, we wanted to prove that these clinical isolates of Trueperella pyogenes were toxic to cells, but we did not compare them with each other, and this comparison was not our goal. We have rewritten the material method and conclusion. We hope this revision can meet the requirements for publication.
Round 3
Reviewer 2 Report
Comments and Suggestions for Authors
The masucript has been sufficiently revised.
Comments on the Quality of English LanguageMinor corrections.